# Multidimensional Fractional Programming for Normalized Cuts

Yannan Chen[1][*]    Beichen Huang[2][*]    Licheng Zhao[3]    Kaiming Shen[1][†]

[1]School of Science and Engineering, The Chinese University of Hong Kong (Shenzhen), China
[2]McMaster University, Canada
[3]Shenzhen Research Institute of Big Data, China
E-mail: yannanchen@link.cuhk.edu.cn, huangb21@mcmaster.ca,
zhaolicheng@sribd.cn, shenkaiming@cuhk.edu.cn

## Abstract

The Normalized cut (NCut) problem is a fundamental and yet notoriously difficult one in the unsupervised clustering field. Because the NCut problem is fractionally structured, the fractional programming (FP) based approach has worked its way into a new frontier. However, the conventional FP techniques are insufficient: the classic Dinkelbach's transform can only deal with a single ratio and hence is limited to the two-class clustering, while the state-of-the-art quadratic transform accounts for multiple ratios but fails to convert the NCut problem to a tractable form. This work advocates a novel extension of the quadratic transform to the multidimensional ratio case, thereby recasting the fractional 0-1 NCut problem into a bipartite matching problem—which can be readily solved in an iterative manner. Furthermore, we explore the connection between the proposed multidimensional FP method and the minorization-maximization theory to verify the convergence.

## 1   Introduction

Fractional programming (FP) is a powerful optimization tool for solving diverse problems involving ratio terms, e.g., in the areas of physics, economics, management science, signal processing, computer science, and information theory [1, 2, 3]. This paper explores a novel application of FP to the normalized cut (NCut)—which is a fundamental and yet notoriously difficult problem for unsupervised data clustering [4]. A new FP technique called the *multidimensional quadratic transform* [5] forms the building block of this work. Differing from the classic Dinkelbach's transform [6] that is typically limited to the single-ratio problem with a pair of scalar-valued numerator and denominator, the multidimensional quadratic transform is capable of handling multiple ratios simultaneously in the same problem, and further accounts for the multidimensional-ratio case wherein the numerators and denominators take a matrix/vector form. It turns out that the NCut problem solving can be made much easier from a multidimensional FP point of view. Two main results have been achieved under the umbrella of FP. First, we show that one most recent advance [7] in the NCut field can be interpreted as a special scalar-ratio version of the multidimensional quadratic transform, which already outperforms the classic methods significantly. Second, by fully exploiting the multidimensional quadratic transform [5], we develop a superior FP-based algorithm tailored to the NCut problem.

Clustering has been considered extensively in the literature from a variety of perspectives, e.g., K-means [8], hierarchical clustering [9], spectral clustering (SC) [10], graph cuts [11], and high-density clustering [12]. The graph cuts approach is of particular interest for its flexibility to cope with a wide range of cluster types, e.g., not requiring the desired clusters to be center-based as many other

---

[*]Equal contribution. Codes available at `https://github.com/zhanchendao/FPC`.
[†]Corresponding author.

38th Conference on Neural Information Processing Systems (NeurIPS 2024).

geometry-based clustering algorithms do [13]. With each data point mapped to a vertex in a weighted undirected graph, there are different ways to measure the relative strength of similarities between subgraphs (each corresponding to a cluster), which in turn lead to different classes of graph cuts algorithms, e.g., min cut [14], ratio cut [15], and min-max cut [16] aside from the NCut. Like many modern works in the realm of graph cuts, our study focuses on the NCut metric because it yields stable performance and prevents cluster imbalance [4].

Nevertheless, the optimization criterion of the NCut is numerically difficult to tackle. To be more specific, the NCut entails solving an NP-complete problem [4]. The SC method constitutes a popular heuristic approach to the NCut problem [4, 10], but it cannot provide any performance guarantee. Other works aim at the analytical aspect and rely heavily on the optimization theory. For example, the Fast Coordinate Descent (FCD) algorithm proposed in [17] evolves from the standard optimization tool of block coordinate descent. By contrast, the Direct Normalized Cut (DNC) algorithm in [18] is somewhat less straightforward. The main idea of [18] is to approximate the NCut problem by using a lower bound on the original optimization objective, but it incurs a costly inner iteration in computing such a lower bound. To remedy this, the Fast Iterative Normalized Cut (FINC) in [7] approximates the NCut problem based on a closed-form lower bound. However, the resulting new problem is still difficult to solve directly, which can only be addressed in a heuristic fashion as shown in [7]. The present work is most closely related to DNC [18] and FINC [7] in the sense that it seeks to approximate the NCut problem via bounding as well. As compared to the above existing bounds, the new bound proposed in this work can be constructed immediately, and can further enable efficient solving of the new problem for the clustering purpose.

## 2 NCut problem statement

Suppose there are $N$ data points in total. Use $i, j \in \{1, 2, \ldots, N\}$ to index these data points. For a pair of data points $i$ and $j$, the similarity between them is quantified as $0 \leq w_{ij} \leq 1$. By symmetry, we have $w_{ij} = w_{ji}$. In the graph theory context, with each data point visualized as a vertex, the edge between vertex $i$ and vertex $j$ is assigned the weight $w_{ij}$ (or $w_{ji}$). Denote by $\mathcal{V}$ the set of vertices, and $\mathcal{E}$ the set of edges. The resulting graph $G = (\mathcal{V}, \mathcal{E})$ can be recognized as a weighted undirected graph. For each vertex $i$, its *degree* $d_i$ is the sum weights across all the incident edges:

$$d_i = \sum_{j=1}^{N} w_{ij}. \tag{1}$$

Dividing the $N$ data points into $K > 1$ clusters is equivalent to partitioning $\mathcal{V}$ into $K$ disjoint subsets $\{\mathcal{V}_1, \mathcal{V}_2, \ldots, \mathcal{V}_K\}$, where $\bigcup_{k=1}^{K} \mathcal{V}_k = \mathcal{V}$ and $\mathcal{V}_k \cap \mathcal{V}_{k'} = \emptyset$ for any $k \neq k'$. For any two disjoint subsets $\mathcal{A}, \mathcal{B} \subseteq \mathcal{V}$, we define

$$\Phi(\mathcal{A}, \mathcal{B}) = \sum_{i \in \mathcal{A}} \sum_{j \in \mathcal{B}} w_{ij}, \tag{2}$$

which is illustrated in Figure 1.

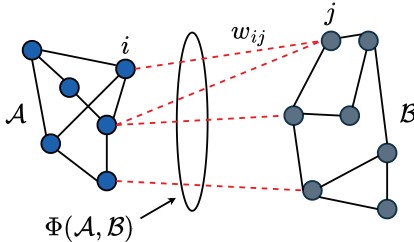

Figure 1: Graph cut between two disjoint subsets $\mathcal{A}$ and $\mathcal{B}$.

Moreover, for any subset $\mathcal{A} \subseteq \mathcal{V}$, we define its *volume* to be

$$\mathrm{vol}(\mathcal{A}) = \sum_{i \in \mathcal{A}} d_i. \tag{3}$$

In principle, data clustering aims to group together those data points that are sufficiently similar to each other. Equivalently, we wish to minimize the similarity between any two clusters. Toward this end, one traditional strategy is to minimize

$$\mathrm{cut}(\mathcal{V}_1, \mathcal{V}_2, \ldots, \mathcal{V}_K) = \frac{1}{2} \sum_{k=1}^{K} \Phi(\mathcal{V}_k, \bar{\mathcal{V}}_k), \tag{4}$$

where $\bar{\mathcal{V}}_k$ is the complement of $\mathcal{V}_k$, i.e., $\bar{\mathcal{V}}_k = \mathcal{V} \backslash \mathcal{V}_k$. However, minimizing $\mathrm{cut}(\mathcal{V}_1, \mathcal{V}_2, \ldots, \mathcal{V}_K)$ alone can be problematic—it tends to put most data points in one particular cluster while leaving other clusters almost empty, namely the cluster imbalance [19]. To resolve this issue, a natural idea is to regularize the cluster volume by considering the normalized cut:

$$\mathrm{ncut}(\mathcal{V}_1, \mathcal{V}_2, \ldots, \mathcal{V}_K) = \frac{1}{2} \sum_{k=1}^{K} \frac{\Phi(\mathcal{V}_k, \bar{\mathcal{V}}_k)}{\mathrm{vol}(\mathcal{V}_k)}. \tag{5}$$

Intuitively speaking, the value of $\Phi(\mathcal{V}_k, \bar{\mathcal{V}}_k)$ would soar if very few data points have been assigned to cluster $k$, thereby discouraging the cluster imbalance.

We are now ready to formalize the NCut problem. The indicator variable $x_{ik} \in \{0, 1\}$ equals 1 if data point $i$ is assigned to cluster $k$, and equals 0 otherwise. Moreover, write $\boldsymbol{W} = [w_{ij}] \in \mathbb{R}^{N \times N}$, $\boldsymbol{D} = \mathrm{diag}[d_1, d_2, \ldots, d_N] \in \mathbb{R}^{N \times N}$, and $\boldsymbol{X} = [x_{ik}] \in \{0, 1\}^{N \times K}$. Denote by $\boldsymbol{x}_k \in \{0, 1\}^N$ the $k$th column of $\boldsymbol{X}$. It can be shown that

$$\mathrm{ncut}(\mathcal{V}_1, \mathcal{V}_2, \ldots, \mathcal{V}_K) = \frac{1}{2} \sum_{k=1}^{K} \frac{\boldsymbol{x}_k^\top \boldsymbol{L} \boldsymbol{x}_k}{\boldsymbol{x}_k^\top \boldsymbol{D} \boldsymbol{x}_k}, \tag{6}$$

where the *graph Laplacian matrix* $\boldsymbol{L}$ is given by $\boldsymbol{L} = \boldsymbol{D} - \boldsymbol{W}$. We seek the optimal clustering decision $\boldsymbol{X}$ that minimizes $\mathrm{ncut}(\mathcal{V}_1, \mathcal{V}_2, \ldots, \mathcal{V}_K)$. Further, because $\mathrm{ncut}(\mathcal{V}_1, \mathcal{V}_2, \ldots, \mathcal{V}_K) = \frac{1}{2} K - \frac{1}{2} \sum_{k=1}^{K} \frac{\boldsymbol{x}_k^\top \boldsymbol{W} \boldsymbol{x}_k}{\boldsymbol{x}_k^\top \boldsymbol{D} \boldsymbol{x}_k}$, the NCut minimization problem boils down to

$$\underset{\boldsymbol{X}}{\mathrm{maximize}} \quad \sum_{k=1}^{K} \frac{\boldsymbol{x}_k^\top \boldsymbol{W} \boldsymbol{x}_k}{\boldsymbol{x}_k^\top \boldsymbol{D} \boldsymbol{x}_k} \tag{7a}$$

$$\mathrm{subject\ to} \quad \sum_{k=1}^{K} x_{ik} = 1, \quad i = 1, \ldots, n \tag{7b}$$

$$x_{ik} \in \{0, 1\}, \quad i = 1 \ldots, n, \ k = 1, \ldots, K, \tag{7c}$$

where the two constraints (7b) and (7c) state that each data point must be assigned to one unique cluster. The difficulties of the above problem can be recognized with two respects. First, the clustering variables $\{x_{ik}\}$ are discrete. Second, even when every $x_{ik}$ is relaxed to be a continuous variable on the interval $[0, 1]$, the problem is still nonconvex.

## 3 Fractional programming

The NCut problem in (7) is fractionally structured. To be more specific, (7) takes a sum-of-ratios form. This quick observation strongly suggests that the NCut is amenable to FP, but it turns out that very few previous works in the literature have adopted the FP approach. In the rest of this section, we first review the conventional FP methods to show why they are rarely considered for the NCut, and then introduce a recently proposed FP technique called the multidimensional quadratic transform—which forms the building block of our proposed clustering algorithm as introduced in Section 4.

### 3.1 Conventional FP methods

The early studies in the FP field are restricted to the *single-ratio* problem:

$$\underset{x \in \mathcal{X}}{\mathrm{maximize}} \quad \frac{A(x)}{B(x)}, \tag{8}$$

where $A(x) \geq 0$ is a nonnegative function, $B(x) > 0$ is a strictly positive function, and $\mathcal{X}$ is a nonempty constraint set on $x$. In the literature, many works further assume that $A(x)$ is concave in $x$, $B(x)$ is convex in $x$, and $\mathcal{X}$ is a convex set, namely the *concave-convex condition*. Notice that problem (8) is still nonconvex in general under the concave-convex condition, so the direct solving of (8) is difficult. The classic Dinkelbach's transform in essence aims to decouple the ratio:

**Proposition 1 (Dinkelbach's transform [6])** *The single-ratio problem* (8) *is equivalent to*

$$\underset{x \in \mathcal{X}}{maximize} \quad A(x) - yB(x), \tag{9}$$

*where the auxiliary variable $y$ is iteratively updated as $y = A(x)/B(x)$.*

Observe that the new problem (9) is convex in $x$ for fixed $y$ under the concave-convex condition, and hence can be efficiently solved by the standard optimization method. Importantly, solving for $x$ in (9) with $y$ iteratively updated guarantees convergence to the global optimum of the original problem (8). However, it is difficult to extend Dinkelbach's transform to the multi-ratio problems (except for the max-min-ratios case [20]). As such, the use of Dinkelbach's transform in the NCut area is limited to the two-class clustering that only needs to optimize a single ratio [21].

We now consider $K > 1$ pairs of the numerator function $A_k(x) \geq 0$ and denominator function $B_k(x) > 0$ along with a nonempty constraint set $\mathcal{X}$. A *sum-of-ratios* problem is then formulated as

$$\underset{x \in \mathcal{X}}{maximize} \quad \sum_{k=1}^{K} \frac{A_k(x)}{B_k(x)}. \tag{10}$$

It is tempting to decouple each ratio $A_k(x)/B_k(x)$ by using Dinkelbach's transform separately, but the resulting new problem is not equivalent to problem (10). Consequently, the classic Dinkelbach's transform does not work for the NCut with general $K$ clusters. A valid method to decouple multiple ratios is presented in the following proposition.

**Proposition 2 (Quadratic transform [5])** *The sum-of-ratios problem* (10) *is equivalent to*

$$\underset{x \in \mathcal{X}, \, y_k \in \mathbb{R}}{maximize} \quad \sum_{k=1}^{K} 2y_k \sqrt{A_k}(x) - y_k^2 B_k(x), \tag{11}$$

*in the sense that $x^\star$ is a solution to* (10) *if and only if $(x^\star, y^\star)$ is a solution to* (11)*, where an auxiliary variable $y_k$ is introduced for each ratio term $A_k(x)/B_k(x)$.*

We propose optimizing $x$ and $\{y_k\}$ iteratively. When $x$ is held fixed, each $y_k$ can be optimally determined as

$$y_k^\star = \frac{\sqrt{A_k(x)}}{B_k(x)}. \tag{12}$$

Furthermore, under a *generalized concave-convex condition* [5] wherein each $A_k(x)$ is a concave function, each $B_k(x)$ is a convex function, and $\mathcal{X}$ is a convex set, it can be shown that the new problem (11) is convex in $x$ when $\{y_k\}$ are held fixed. Thus, the alternating optimization between $x$ and $\{y_k\}$ can be performed efficiently.

Now let us return to the NCut problem in (7) and apply the above FP technique to it. Treating $\boldsymbol{x}_k^\top \boldsymbol{W} \boldsymbol{x}_k$ and $\boldsymbol{x}_k^\top \boldsymbol{D} \boldsymbol{x}_k$ respectively as numerator and denominator, we can recast problem (7) into

$$\underset{\boldsymbol{X}, \, y_k \in \mathbb{R}}{maximize} \quad \sum_{k=1}^{K} \left( 2y_k \sqrt{\boldsymbol{x}_k^\top \boldsymbol{W} \boldsymbol{x}_k} - y_k^2 \boldsymbol{x}_k^\top \boldsymbol{D} \boldsymbol{x}_k \right) \tag{13a}$$

$$\text{subject to} \quad (7b), \ (7c). \tag{13b}$$

As before, we optimize $\boldsymbol{X}$ and $\{y_k\}$ iteratively. For fixed $\boldsymbol{X}$, the optimal solution of $y_k$ is

$$y_k^\star = \frac{\sqrt{\boldsymbol{x}_k^\top \boldsymbol{W} \boldsymbol{x}_k}}{\boldsymbol{x}_k^\top \boldsymbol{D} \boldsymbol{x}_k}. \tag{14}$$

It remains to optimize $\boldsymbol{X}$ in (13) for fixed $\{y_k\}$. Observe that $\boldsymbol{x}^\top \boldsymbol{W}\boldsymbol{x}$ is not a concave function of $\boldsymbol{x}$ since $\boldsymbol{W}$ is often a positive semi-definite matrix [22], and also that the constraint set of $\boldsymbol{X}$ is not convex because of (7c), so the aforementioned generalized concave-convex condition does not hold here. As a result, solving for $\boldsymbol{X}$ in (13) with $\{y_k\}$ held fixed is no longer a convex problem. The above alternating optimization between $\boldsymbol{X}$ and $\{y_k\}$ can be recognized as the so-called *Fast Iterative Normalized Cut (FINC)* algorithm of the recent work [7]. Although it manages to decouple multiple ratios in the NCut problem, we are faced with a new challenging problem. The new problem is dealt with in a heuristic fashion in [7]. This fact perhaps explains why the FP approach has not yet been considered extensively in the literature despite the fractional structure of the NCut problem.

## 3.2 Multidimensional FP method

We now proceed to a much more sophisticated FP toolkit that accounts for multidimensional ratios. To start, consider the following matrix extension of the traditional scalar-valued FP problem: each $A_k(x) \geq 0$ is generalized as positive semi-definite $\boldsymbol{A}_k(x) \in \mathbb{S}_+^{m \times m}$, while each $B_k(x) > 0$ is generalized as positive definite $\boldsymbol{B}_k(x) \in \mathbb{S}_{++}^{m \times m}$. Accordingly, the ratio term is extended to the matrix form as

$$\frac{A_k(x)}{B_k(x)} \in \mathbb{R}_+ \implies \boldsymbol{B}_k(x)^{-1}\boldsymbol{A}_k(x) \in \mathbb{S}_+^{m \times m}.$$

We then arrive at a matrix extension of the sum-of-ratios problem (10):

$$\underset{x \in \mathcal{X}}{\text{maximzie}} \quad \sum_{k=1}^{K} \text{tr}\left(\boldsymbol{B}_k^{-1}(x)\boldsymbol{A}_k(x)\right). \tag{15}$$

One main result of this paper is that the quadratic transform in Proposition 2 carries over to the matrix ratio case, as stated in the following proposition.

**Proposition 3 (Multidimensional quadratic transform)** *Suppose that each $\boldsymbol{A}_k(x) \in \mathbb{S}_+^{m \times m}$ can be factorized as*

$$\boldsymbol{A}_k(x) = [\boldsymbol{Z}_k(x)]^\top [\boldsymbol{Z}_k(x)] \quad \text{where} \quad \boldsymbol{Z}_k(x) \in \mathbb{R}^{\ell \times m} \tag{16}$$

*for some positive integer $\ell$. The matrix FP problem* (15) *is then equivalent to*

$$\underset{x \in \mathcal{X},\, \boldsymbol{Y}_k \in \mathbb{R}^{\ell \times m}}{\text{maximize}} \quad \sum_{k=1}^{K} \text{tr}\left(2\boldsymbol{Y}_k[\boldsymbol{Z}_k(x)]^\top - \boldsymbol{Y}_k\boldsymbol{B}_k(x)\boldsymbol{Y}_k^\top\right), \tag{17}$$

*where an auxiliary variable $\boldsymbol{Y}_k \in \mathbb{R}^{\ell \times m}$ is introduced for each matrix ratio $\boldsymbol{B}_k^{-1}(x)\boldsymbol{A}_k(x)$.*

**Proof 1** *It can be shown that each $\boldsymbol{Y}_k$ in* (17) *is always optimally determined as*

$$\boldsymbol{Y}_k^\star = \boldsymbol{Z}_k(x)\boldsymbol{B}_k^{-1}(x). \tag{18}$$

*Substituting the above $\boldsymbol{Y}_k^\star$ in* (17) *recovers the original problem* (15).

**Proposition 4** *The alternating optimization between $x$ and $\{\boldsymbol{Y}_k\}$ in* (17) *amounts to an MM procedure, so it guarantees a nondecreasing convergence of the original optimization objective in* (15)*, as specified in Appendix A.1.*

The key step is to optimize $x$ for fixed $\{\boldsymbol{Y}_k\}$. Recall that the primal variable $x$ is still difficult to optimize for the NCut problem after applying the quadratic transform in Proposition 2. In contrast, it turns out that the multidimensional quadratic transform in Proposition 3 can lead us to an efficient iterative update of $x$ for the NCut problem scenario, as elaborated in the next section.

## 4 Proposed Multidimensional-FP-based NCut

The goal of this section is to address problem (7) by means of the multidimensional FP. We begin with a special case in which the similarity matrix $\boldsymbol{W}$ is assumed to be positive semi-definite; the indefinite $\boldsymbol{W}$ case will be discussed later on.

It is crucial to notice that the numerator part $x_k^\top W x_k$ can be factorized as

$$x_k^\top W x_k = z_k^\top z_k \quad \text{where} \quad z_k = W^{\frac{1}{2}} x_k. \tag{19}$$

We remark that $W^{\frac{1}{2}}$, i.e., the symmetric square root of $W$ [23], is guaranteed to exist because we have assumed that $W \in \mathbb{S}_+^{m \times m}$ for the current discussion. We then treat $x_k^\top W x_k$, $z_k$, and $x_k^\top D x_k$ as $A_k(x)$, $Z_k(x)$, and $B_k(x)$, respectively, in Proposition 3, with $m = 1$ and $\ell = N$, thus using the multidimensional quadratic transform to reformulate the NCut problem (7) as

$$\underset{X, y_k \in \mathbb{R}^N}{\text{maximize}} \quad \sum_{k=1}^K \text{tr}\left(2 y_k (W^{\frac{1}{2}} x_k)^\top - y_k (x_k^\top D x_k) y_k^\top\right) \tag{20a}$$

$$\text{subject to} \quad \text{(7b), (7c).} \tag{20b}$$

We then optimize $X$ and $\{y_k\}$ iteratively. When $X$ is held fixed, each $y_k$ is optimally determined as

$$y_k^\star = \frac{W^{\frac{1}{2}} x_k}{x_k^\top D x_k}. \tag{21}$$

Now the core question is whether $X$ could be efficiently solved when $\{y_k\}$ are held fixed. The optimization objective of $X$ in (20a) for fixed $\{y_k\}$ is written as

$$h(X) = \sum_{k=1}^K \text{tr}\left(2 y_k (W^{\frac{1}{2}} x_k)^\top - y_k (x_k^\top D x_k) y_k^\top\right). \tag{22}$$

It is critical to observe that under the discrete constraint $x_{ik} \in \{0, 1\}$ we must have

$$x_k^\top D x_k = \mathbf{1}^\top D x_k = \delta^\top x_k, \tag{23}$$

where $\mathbf{1} = (1, 1, \ldots, 1)^\top$ is the all-ones vector and

$$\delta = \mathbf{1}^\top D = [d_1, d_2, \ldots, d_N]^\top. \tag{24}$$

We can then rewrite $h(X)$ as

$$h(X) = \sum_{k=1}^K \left(2 y_k^\top W^{\frac{1}{2}} x_k - y_k^\top y_k \delta^\top x_k\right) = \sum_{k=1}^K \mu_k^\top x_k, \tag{25}$$

where

$$\mu_k = 2 W^{\frac{1}{2}} y_k - \delta y_k^\top y_k. \tag{26}$$

In (25), the first equality follows since $x_k^\top D x_k$ is a scalar. Denote by $\mu_{ik}$ the $i$th component of $\mu_k$. In light of (25), we can readily maximize $h(X)$ under the constraints (7b) and (7c): it is optimal to set $x_{ik}$ with the largest $\mu_{ik}$ on each row of $X$ to one, while setting the rest $x_{ik}$ of the row to zero, i.e.,

$$x_{ik}^\star = \begin{cases} 1 & \text{if } k = \underset{k'}{\arg\max} \; \mu_{ik'} \\ 0 & \text{otherwise.} \end{cases} \tag{27}$$

If there exists a tie (i.e., when more than one cluster index $k'$ maximizes $\mu_{ik'}$ for same $i$) then break it randomly. Further, with $y_k^\star$ in (21) plugged in (26), we obtain an efficient computation of $\mu_k$ as

$$\mu_k = \frac{2 W x_k}{x_k^\top D x_k} - \frac{\delta x_k^\top W x_k}{(x_k^\top D x_k)^2} = \frac{2 W x_k}{\delta^\top x_k} - \frac{\delta x_k^\top W x_k}{(\delta^\top x_k)^2}. \tag{28}$$

The merits of rewriting $\mu_k$ as (28) are two-fold. First, it sidesteps the update of the auxiliary variables $\{y_k\}$. Second, it no longer entails computing the square root of $W$. The resulting algorithm referred to as *fractional programming-based clustering (FPC)* is summarized in the following.

Clearly, the FPC algorithm is guaranteed to converge in terms of the new objective value $h(X)$, since the iterative update of $X$ and $\{\mu_k\}$ in FPC amounts to a block coordinate ascent for problem (20) so that $h(X)$ is monotonically increasing after each iteration. We can actually claim a stronger result for FPC according to Proposition 4, as stated in the subsequent proposition.

---

**Algorithm 1** Proposed fractional programming-based clustering (FPC)

---

1: Initialize $\boldsymbol{X}$ to a feasible value satisfying $\sum_{k=1}^{K} x_{ik} = 1$ and $x_{ik} \in \{0, 1\}$ for each $(i, k)$.
2: **repeat**
3:     Update each $\boldsymbol{\mu}_k$ according to (28), for $k = 1, 2, \ldots, K$.
4:     Update each $x_{ik}$ according to (27), for $i = 1, 2, \ldots, N$ and $k = 1, 2, \ldots, K$.
5: **until** the value of $h(\boldsymbol{X}) = \sum_{k=1}^{K} \boldsymbol{\mu}_k^\top \boldsymbol{x}_k$ converges

---

**Proposition 5** *Not only the new objective value $h(\boldsymbol{X})$ in (22) but also the original objective value of the NCut problem, $f(\boldsymbol{X}) = \sum_{k=1}^{K} \frac{\boldsymbol{x}_k^\top \boldsymbol{W} \boldsymbol{x}_k}{\boldsymbol{x}_k^\top \boldsymbol{D} \boldsymbol{x}_k}$, is nondecreasing after each iteration of FPC.*

We thus far assume that the similarity matrix $\boldsymbol{W}$ is positive semi-definite; this assumption can be justified by arguing that a positive definite kernel [24] (e.g., the Gaussian kernel) is often used to generate $\boldsymbol{W}$. But what if some indefinite kernel has been adopted and hence $\boldsymbol{W}$ is not necessarily positive semi-definite anymore? The following proposition provides a solution.

**Proposition 6** *Suppose that the similarity matrix $\boldsymbol{W}$ is indefinite. We can choose a sufficiently large $\alpha > 0$ so that the new matrix*

$$\widetilde{\boldsymbol{W}} = \boldsymbol{W} + \alpha \boldsymbol{D} \tag{29}$$

*is positive semi-definite. Notice that such $\alpha$ must exist since*

$$\alpha = -\frac{\lambda_{\min}(\boldsymbol{W})}{\min_i d_i} \tag{30}$$

*is a feasible choice. Then we can equivalently consider problem (7) with $\widetilde{\boldsymbol{W}}$ used in place of $\boldsymbol{W}$, which can be readily addressed by the FPC algorithm.*

**Proof 2** *See Appendix A.2.*

Finally, we examine the computational complexity of the FPC algorithm. The update of $\boldsymbol{X}$ as in (27) incurs a computational complexity of $\mathcal{O}(K^2 N)$, while the update of $\{\boldsymbol{\mu}_k\}$ as in (28) incurs a computational complexity of $\mathcal{O}(K N^2)$. If $\boldsymbol{W}$ is indefinite, then we would further find the smallest eigenvalue of $\boldsymbol{W}$ as required in (30). Rather than computing all the eigenvalues and then picking the smallest, which incurs $\mathcal{O}(N^3)$, we propose a more efficient way of computing $\lambda_{\min}$:

1. Find the largest eigenvalue of $\|\boldsymbol{W}\|_{\mathrm{F}} \boldsymbol{I} - \boldsymbol{W}$, denoted as $\lambda_1$, by the power method [25], where $\| \cdot \|_{\mathrm{F}}$ is the Frobenius norm.

2. Compute the smallest eigenvalue as $\lambda_{\min}(\boldsymbol{W}) = \|\boldsymbol{W}\|_{\mathrm{F}} - \lambda_1$.

Thus, the overall complexity of finding $\alpha$ as in (30) is $\mathcal{O}(N^2)$. To sum up, the per-iteration complexity of FPC equals $\mathcal{O}(K N^2)$, while the traditional SC algorithm incurs a complexity of $O(N^3)$.

## 5 Experiments

We validate the performance of the proposed FPC algorithm on 8 common datasets as summarized in Table 1. The benchmarks are the SC [4], FINC [7], and FCD [17]. We use the Gaussian kernel to generate the similarity matrix, i.e., $w_{ij} = \exp\left(-\|\boldsymbol{v}_i - \boldsymbol{v}_j\|_2^2\right)$, where $\boldsymbol{v}_i$ and $\boldsymbol{v}_j$ are the feature vectors of data points $i$ and $j$. All the tests were carried out on a desktop equipped with 2.10 GHz CPU$\times$12. Throughout the tables, we highlight the best performance by using the bold font.

### 5.1 Optimization objective of NCut

We first evaluate the performance of the different algorithms in minimizing $\mathrm{ncut}(\mathcal{V}_1, \mathcal{V}_2, \ldots, \mathcal{V}_K)$ as defined in (6). We run each algorithm 10 times with the random starting point generated for each trial, and then pick the best one. Table 2 summarizes the results. Observe that the proposed FPC method achieves the lowest NCut objective across all the datasets. For instance, the NCut objective of FPC is 0.19% lower than that of FCD for the dataset Office+Caltech10 with $K = 10$ clusters, and

Table 1: Datasets used for the task of dividing $N$ data points into $K$ clusters.

| Dataset | $N$ | $K$ | Number of features | Source |
|---|---|---|---|---|
| Breast | 106 | 6 | 9 | UCI datasets[26] |
| Thyroid | 215 | 3 | 5 | UCI datasets[26] |
| Office+Caltech10 | 2533 | 10 | 800 | Github transfer-learning[27] |
| Splice | 3175 | 3 | 240 | UCI datasets[26] |
| Rice | 3810 | 2 | 7 | UCI datasets[26] |
| Landsat | 6435 | 7 | 36 | UCI datasets[26] |
| USPS | 9298 | 10 | 256 | LIBSVM[28] |
| Epileptic | 11500 | 5 | 178 | UCI datasets[26] |

Table 2: NCut objective values achieved by the different algorithms with random initialization.

| | SC | FINC | FCD | FPC |
|---|---|---|---|---|
| Breast | 2.438568 | 2.445278 | 2.446499 | **2.431813** |
| Thyroid | 0.983144 | 0.983393 | 0.986163 | **0.983115** |
| Office+Caltech10 | 4.483921 | 4.483962 | 4.491925 | **4.483501** |
| Splice | 0.997651 | 0.999867 | 0.998569 | **0.997636** |
| Rice | **0.499193** | 0.499999 | 0.499209 | **0.499193** |
| Landsat | 2.994675 | 2.999986 | 2.995302 | **2.994335** |
| USPS | 4.476479 | 4.476404 | 4.476591 | **4.475869** |
| Epileptic | 1.992376 | 1.991369 | 1.992688 | **1.991311** |

Table 3: NCut objective values achieved by the different algorithms with the SC initialization.

| | SC | SC+FINC | SC+FCD | SC+FPC |
|---|---|---|---|---|
| Breast | 2.438695±7.5e-5 | 2.442353±4.1e-5 | 2.438695±7.5e-5 | **2.437931±2.8e-4** |
| Thyroid | **0.983144±0.0** | 0.989329±0.0 | **0.983144±0.0** | **0.983144±0.0** |
| Office+Caltech10 | 4.483945±1.1e-5 | 4.483373±6.0e-6 | 4.483635±2.9e-5 | **4.483280±9.0e-6** |
| Splice | 0.997651±0.0 | 0.997651±0.0 | 0.997651±2.0e-5 | **0.997638±0.0** |
| Rice | **0.499193±0.0** | **0.499193±0.0** | **0.499193±0.0** | **0.499193±0.0** |
| Landsat | 2.994678±2.0e-6 | 2.994678±2.0e-6 | 2.994499±7.5e-5 | **2.994335±0.0** |
| USPS | 4.476546±1.7e-4 | 4.475926±1.4e-4 | 4.475932±1.4e-4 | **4.475913±1.3e-4** |
| Epileptic | 1.992378±3.0e-6 | 1.991756±1.5e-5 | 1.991353±0.0 | **1.991313±0.0** |

Note: Each entry has the form [objective value]±[standard variance]. Red color indicates degradation while blue color indicates improvement.

0.07% lower for the dataset Epileptic with $N = 11500$ data points. All the benchmarks except SC are strictly inferior to FPC; SC is equally good as FPC only on the dataset Rice.

Moreover, we consider first using SC to obtain a raw clustering decision and then using other algorithms to refine it. The test results are summarized in Table 3. Observe that using FPC after SC can achieve the best performance on all 8 datasets. In particular, it strictly improves upon the SC initialization on 6 datasets. It is worth observing that FINC may even yield worse performance after the initialization by SC; this is because FINC cannot guarantee that the new problem is optimally solved per iteration as formerly mentioned in Section 3.1. Finally, in Fig. 2 we find the global optimum for two small-size datasets via exhaustive search, and use it as the benchmark to compare with the proposed FPC algorithm; observe that FPC attains convergence to the global optimum after merely 3 iterates.

## 5.2 Other performance metrics

Aside from the Ncut optimization criterion, the following commonly used performance metrics in practice are considered for the different clustering algorithms: the accuracy (ACC), the normalized mutual information (NMI) [29], and the adjusted random index (ARI) [30]. Unlike $\mathrm{ncut}(\mathcal{V}_1, \mathcal{V}_2, \ldots, \mathcal{V}_K)$, the above metrics are proportional to the performance, i.e., the higher metric value, the better clustering. The test results are summarized in Table 4. Although these performance metrics are not directly tied to the NCut objective, the proposed FPC method still achieves the highest scores in many cases.

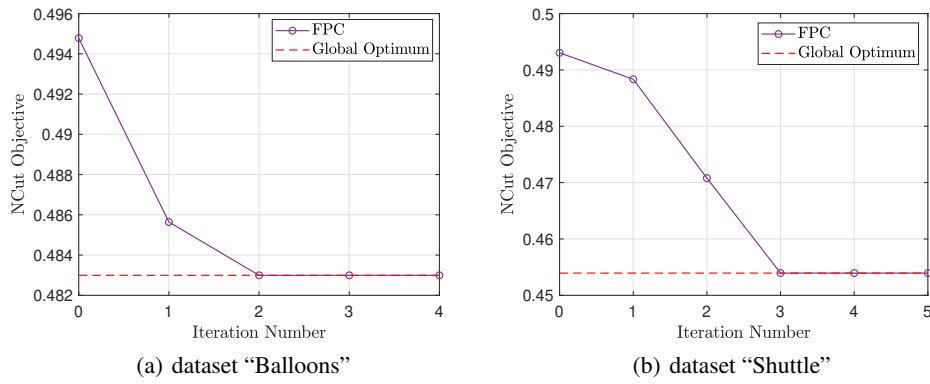

|   | (a) dataset "Balloons" | (b) dataset "Shuttle" |
|---|---|---|

Figure 2: Convergence in terms of the NCut objective value for two UCI datasets.

Table 4: The performance metrics of ACC, NMI, and ARI achieved by the different algorithms.

|  | SC | FINC | FCD | FPC | SC | FINC | FCD | FPC |
|---|---|---|---|---|---|---|---|---|
| Dataset | | Breast | | | | Thyroid | | |
| ACC | 0.4906 | 0.5000 | 0.3868 | **0.5283** | 0.8837 | **0.9163** | 0.7860 | 0.9070 |
| NMI | 0.4942 | 0.4798 | 0.3525 | **0.5052** | 0.4956 | **0.6061** | 0.3396 | 0.5780 |
| ARI | 0.2931 | 0.3104 | 0.1582 | **0.3379** | 0.6082 | **0.7167** | 0.4121 | 0.6869 |
| Dataset | | Office+Caltech10 | | | | Splice | | |
| ACC | 0.3533 | 0.3308 | 0.1717 | **0.3640** | 0.6646 | 0.3512 | 0.5096 | **0.7225** |
| NMI | 0.2347 | 0.2358 | 0.0704 | **0.2491** | 0.2856 | 0.0015 | 0.2149 | **0.3529** |
| ARI | 0.1440 | 0.1314 | 0.0440 | **0.1469** | 0.2686 | 0.0007 | 0.1650 | **0.3514** |
| Dataset | | Rice | | | | Landsat | | |
| ACC | 0.8966 | 0.5142 | 0.8853 | **0.8992** | 0.6044 | 0.1584 | 0.5809 | **0.6611** |
| NMI | 0.5129 | 0.0006 | 0.4845 | **0.5216** | 0.4905 | 0.0013 | 0.4073 | **0.6111** |
| ARI | 0.6288 | 0.0005 | 0.5937 | **0.6371** | 0.3948 | 0.0000 | 0.3423 | **0.5328** |
| Dataset | | USPS | | | | Epileptic | | |
| ACC | 0.7037 | 0.6861 | **0.7060** | 0.6920 | 0.3609 | 0.3386 | **0.3743** | 0.3197 |
| NMI | **0.6419** | 0.6406 | 0.6333 | 0.6377 | 0.1627 | **0.2419** | 0.1909 | 0.2370 |
| ARI | 0.5724 | 0.5626 | **0.5734** | 0.5679 | 0.1119 | **0.1385** | 0.1326 | 0.1332 |

We further try out the different clustering algorithms in the image segmentation task. Using the image dataset from [31], we perform the color histogram and the local binary pattern analysis for about 500 superpixels to extract the features as in [32]. As shown in Figure 3, the clustering by FPC gives clearer boundaries of the objects than other methods.

Moreover, Figure 4 shows the average time consumption of the different algorithms. It can be seen that FPC runs 73% faster than FINC, and runs equally fast as SC. We remark that FCD requires the least running time because it tends to get trapped in a suboptimal point prematurely at the early stage.

## 6   Conclusion and limitation

This work proposes a novel application of multidimensional FP to the NCut clustering, differing from the previous works that rely on the traditional scalar-ratio FP such as Dinkelbach's transform and quadratic transform. The main merit of using multidimensional FP is that the new 0-1 problem can be efficiently solved via linear search. Further, the resulting FPC algorithm can be interpreted as an MM procedure with provable monotonic convergence in terms of the NCut optimization criterion. Thus far, we only show that its per-iteration complexity is lower than the overall complexity of the traditional SC algorithm.

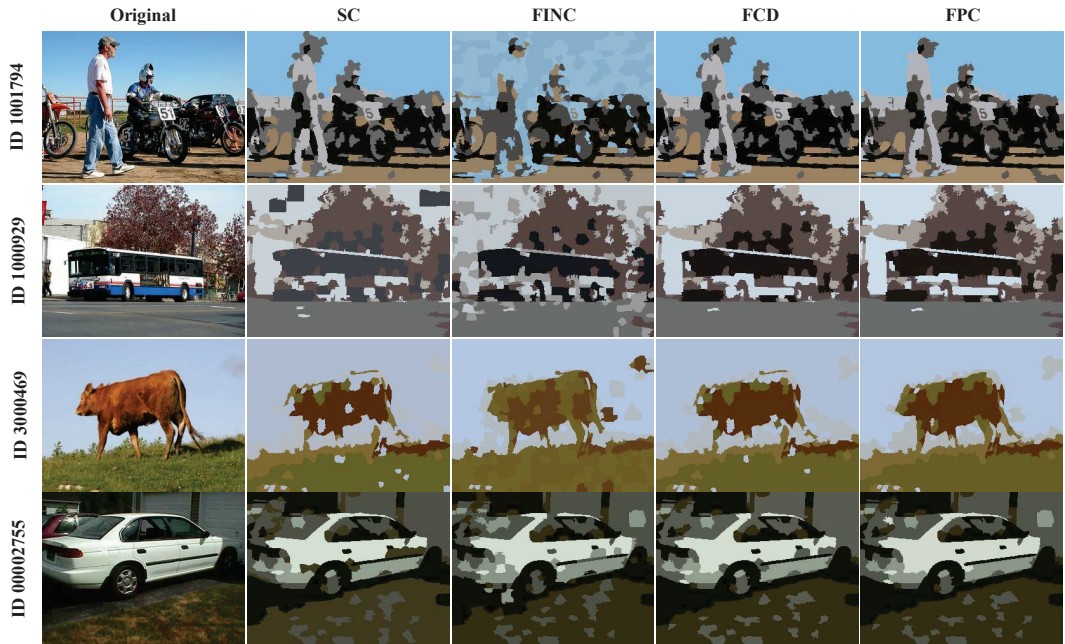

Figure 3: Image segmentation by the different algorithms.

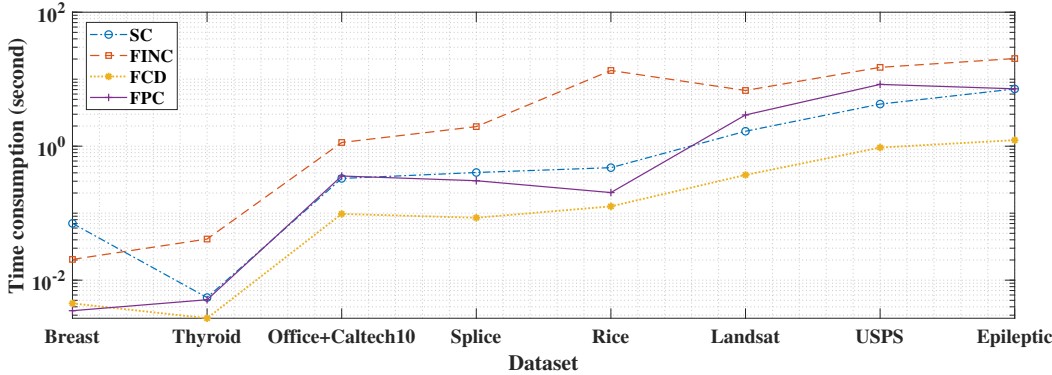

Figure 4: Running time of the different algorithms when applied to the different datasets.

# 7 Acknowledgments

The work of Yannan Chen, Beichen Huang, and Kaiming Shen was supported in part by Guangdong Major Project of Basic and Applied Basic Research (No. 2023B0303000001), in part by the National Natural Science Foundation of China (NSFC) under Grant 92167202, and in part by Shenzhen Steady Funding Program. The work of Licheng Zhao was supported in part by the NSFC under Grant 62206182, and in part by Guangdong Basic and Applied Basic Research Foundation under Grant 2024A1515010154.

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

## Appendix

### A.1 Proof of Proposition 4

We begin with a brief review of the MM theory [33, 34]. Consider a general constrained optimization problem

$$\underset{x \in \mathcal{X}}{\text{maximize}} \quad f(x). \tag{31}$$

Rather than solving the above problem directly, the MM theory deals with an approximation of (31) iteratively. In principle, we construct a *surrogate function* $g(x|\hat{x})$ of $x$ given the condition parameter $\hat{x} \in \mathcal{X}$, such that

$$g(x|\hat{x}) \leq f(x), \tag{32}$$

$$g(\hat{x}|\hat{x}) = f(\hat{x}). \tag{33}$$

Then the MM method solves a sequence of new problems of the surrogate function:

$$\underset{x \in \mathcal{X}}{\text{maximize}} \quad g(x|\hat{x}), \tag{34}$$

with $\hat{x}$ iteratively updated to the previous solution $x$. Specifically, with the solution of (34) in the $(t-1)$th iteration denoted by $\hat{x}^{(t-1)}$, we construct a surrogate function $g(x|\hat{x}^{(t-1)})$ for the $t$ iteration and then obtain the new solution $\hat{x}^{(t)}$ of (34), and so forth, as illustrated in Figure 5.

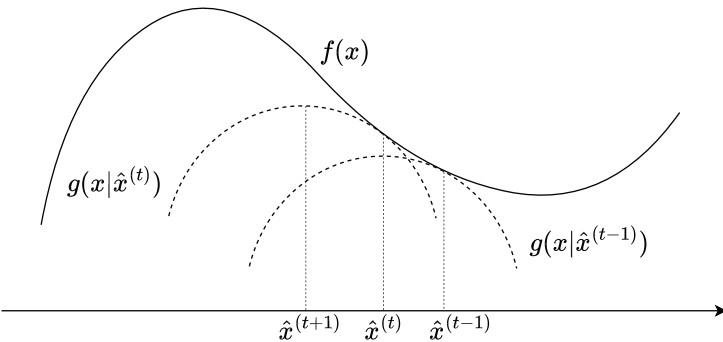

Figure 5: The original objective value $f(x)$ is nondecreasing after each iteration of the MM method.

**Theorem 1 (Monotonic convergence [33, 34])** *The MM method yields a nondecreasing convergence of the original objective value $f(x)$, i.e.,*

$$f(\hat{x}^{(t-1)}) \leq f(\hat{x}^{(t)}) \quad for \quad t = 1, 2, \ldots. \tag{35}$$

We now show that the alternating optimization between $x$ and $\{\boldsymbol{Y}_k\}$ in (17) can be interpreted as an MM method. We still use the superscript $t$ to index the iteration. Recall also that each $\boldsymbol{Y}_k$ in the $t$th iteration is optimally updated as

$$\boldsymbol{Y}_k^{(t)} = \boldsymbol{Z}_k(x^{(t-1)})\boldsymbol{B}_k^{-1}(x^{(t-1)}). \tag{36}$$

We now view the update of $\boldsymbol{Y}_k$ as a function of the previous solution $\hat{x}$, that is

$$\mathcal{Y}_k(\hat{x}) = \boldsymbol{Z}_k(\hat{x}^{(t-1)})\boldsymbol{B}_k^{-1}(\hat{x}^{(t-1)}). \tag{37}$$

Substituting each $\boldsymbol{Y}_k$ with $\mathcal{Y}_k(\hat{x})$ in the new objective function in (17) gives rise to a function of $x$ conditioned on $\hat{x}$:

$$g(x|\hat{x}) = \sum_{k=1}^{K} \text{tr} \left( 2\boldsymbol{Z}_k(\hat{x}^{(t-1)})\boldsymbol{B}_k^{-1}(\hat{x}^{(t-1)})[\boldsymbol{Z}_k(x)]^\top \right.$$
$$\left. - \boldsymbol{Z}_k(\hat{x}^{(t-1)})\boldsymbol{B}_k^{-1}(\hat{x}^{(t-1)})\boldsymbol{B}_k(x)[\boldsymbol{Z}_k(\hat{x}^{(t-1)})\boldsymbol{B}_k^{-1}(\hat{x}^{(t-1)})]^\top \right). \tag{38}$$

Notice that maximizing the objective function in (17) with $\{\boldsymbol{Y}_k\}$ fixed is equivalent to maximizing $g(x|\hat{x})$ where $\hat{x}$ is fixed at the solution of the previous iteration.

Most importantly, it can be shown that $g(x|\hat{x})$ meets the conditions (32) and (33) for the original objective function in (15), so the alternating optimization between $x$ and $\{\boldsymbol{Y}_k\}$ in (17) amounts to an MM procedure. The result of Proposition 4 then immediately follows by Theorem 1.

### A.2 Proof of Proposition 6

First of all, because $\boldsymbol{W}$ is indefinite, its minimum eigenvalue $\lambda_{\min}(\boldsymbol{W})$ must be negative. With $\alpha$ in (30), the matrix $\widetilde{\boldsymbol{W}}$ can be rewritten as

$$
\begin{aligned}
\widetilde{\boldsymbol{W}} &= \boldsymbol{W} - \frac{\lambda_{\min}(\boldsymbol{W})}{\min_i d_i}\boldsymbol{D} \\
&= \boldsymbol{W} - \lambda_{\min}(\boldsymbol{W})\boldsymbol{I} - \lambda_{\min}(\boldsymbol{W})\mathrm{diag}\left[\frac{d_1}{\min_i d_i} - 1, \ldots, \frac{d_N}{\min_i d_i} - 1\right].
\end{aligned} \tag{39}
$$

It is evident that $\boldsymbol{W} - \lambda_{\min}(\boldsymbol{W})\boldsymbol{I}$ and $-\lambda_{\min}(\boldsymbol{W})\mathrm{diag}\left[\frac{d_1}{\min_i d_i} - 1, \ldots, \frac{d_N}{\min_i d_i} - 1\right]$ are both positive semi-definite, so $\widetilde{\boldsymbol{W}}$ is positive semi-definite too. Moreover, it is easy to see that problem (7) is equivalent to

$$
\underset{\boldsymbol{X}}{\text{maximize}} \quad K\alpha + \sum_{k=1}^{K} \frac{\boldsymbol{x}_k^\top \boldsymbol{W} \boldsymbol{x}_k}{\boldsymbol{x}_k^\top \boldsymbol{D} \boldsymbol{x}_k} \tag{40a}
$$

$$
\text{subject to} \quad \text{(7b), (7c),} \tag{40b}
$$

which can be further rewritten as

$$
\underset{\boldsymbol{X}}{\text{maximize}} \quad \sum_{k=1}^{K} \frac{\boldsymbol{x}_k^\top \widetilde{\boldsymbol{W}} \boldsymbol{x}_k}{\boldsymbol{x}_k^\top \boldsymbol{D} \boldsymbol{x}_k} \tag{41a}
$$

$$
\text{subject to} \quad \text{(7b), (7c),} \tag{41b}
$$

The proof is then completed.

