# OpenReview forum: "Multidimensional Fractional Programming for Normalized Cuts"
_NeurIPS.cc/2024/Conference — NeurIPS 2024 poster_

### Official Review · Reviewer_fiXH · 2024-07-05

**Soundness:** 3
**Presentation:** 3
**Contribution:** 2
**Rating:** 6
**Confidence:** 4

**Summary:**

The paper presents a new fractional programming based algorithm for the multi-cluster normalized cut objective which exploits a multidimensional quadratic transform.

The paper starts by introducing the Ncut problem. Then it discusses previous fractional programming methods such as Dinkelbach's method for the single-ratio case, and the quadratic transform that can be used to deal with the sum-of-ratios problem. It is then shown that the quadratic transform carries over to the matrix ratio case. The proposed method FPC works by applying the multidimensional quadratic form to the Ncut objective. They then show that the objective as well as the original Ncut objective are nondecreasing after each iteration of their method FPC.

In the experiments the approach is compared to standard spectral clustering as well as the recently proposed methods FINC and FCD. The proposed method achieves the lowest Ncut value for all datasets, when doing random initializations or refining an existing spectral clustering solution, as well as competitive results when evaluating with other performance metrics. In terms of time consumption, the method runs in the same order of magnitude as standard spectral clustering and faster as FINC method, while being slower than the FCD method.

**Strengths:**

The paper is well-written and technically sound. It proposes a new method for the Ncut objective which has been shown to yield competitive results on several small- to medium-size benchmark datasets. The main contribution of the paper is the application of the multi-dimensional quadratic form to the Ncut objective.

**Weaknesses:**

The used datasets are relatively small, the paper could be made stronger by showing that the method also scales to some larger datasets. Moreover, the connection to the MM theory is mentioned in the last section as a conclusion without being mentioned before. It only gets discussed in the appendix for the first time. The main results should be at least mentioned before in the main body of the text.

**Questions:**

Is Prop. 3 novel? Could you please comment on the differences to Theorem 2 in [5]?

**Limitations:**

The authors mention in the conclusion that the analysis of the convergence speed of the method could still be extended.

---

> ### Author Rebuttal · Authors · 2024-08-06
>
> Thank you so much for appreciating the presentation and the technical contributions of this work. We would like to address your concerns and questions in the following.
>
> 1. **Weakness:** Thanks for the constructive suggestion. We have added two larger datasets:  letter recognition (with 20,000 samples) and MNIST subset (with 30,000 samples). The new experimental results are summarized in Table 1 and Table 2 in the attached one-page PDF. Observe that the proposed FPC algorithm still outperforms the benchmarks significantly on these new datasets. Also thanks for the advice on paper organization. We would move the connection to the MM theory to the main body of the text after the rebuttal session finishes.
> 2. **Questions:** Yes, Prop. 3 is novel. Prop. 3 deals with the matrix ratio
> $$
> \boldsymbol B^{-1}\boldsymbol A\in\mathbb R^{m\times m},
> $$
> while Theorem 2 in [5] deals with the scalar ratio
> $$
>     \boldsymbol a^\top\boldsymbol B^{-1} \boldsymbol a \in\mathbb R.
> $$
> Furthermore, when the ratio is nested in $\mathrm{Tr}(\cdot)$, we rewrite the scalar ratio problem as the matrix ratio case:
> $$
>     \mathrm{Tr}(\boldsymbol a^\top\boldsymbol B^{-1} \boldsymbol a)=\mathrm{Tr}(\boldsymbol B^{-1} \boldsymbol a\boldsymbol a^\top)=\mathrm{Tr}(\boldsymbol B^{-1} \boldsymbol A),\quad\text{where}\quad \boldsymbol A = \boldsymbol a\boldsymbol a^\top.
> $$
> However, the reverse is not true since $\boldsymbol A$ may not be factored as $\boldsymbol a\boldsymbol a^\top$. Thus, Prop. 3 encompasses Theorem 2 in [5] as a special case.
> 3.  **Limitations:** We have accomplished the convergence rate analysis of the proposed FPC algorithm. It can be shown that
> \begin{align*}
>     |f(x^*)-f(x^{(1)})| \le \frac{LR^3}{6},\\
> \end{align*}
> \begin{align}
> |f(x^*)-f(x^{(k)})| \le \frac{2\Lambda R^2+2LR^3/3}{k+3},\quad\text{for}\quad k\ge 2,
> \end{align}
> where $f(\cdot)$ is the optimization objective value, $x^*$ is the converged solution,
> $k$ is the iterate index, $R$ is the Euclidean distance from the starting point to $x^*$, $L$ is the Lipschitz constant of $\nabla^2 f(x)$, $\Lambda$ is the maximum eigenvalue of $\nabla^2 f(x)$, and $x^{(k)}$ is the solution after $k$ iterates. We remark that the convergence rate analysis is highly nontrivial here because the NCut problem is nonconvex and incurs discrete constraints. The above analysis will be added to the paper as a major theoretic contribution.

---

> > ### Comment · Reviewer_fiXH · 2024-08-12
> > **Reponse to Rebuttal**
> >
> > In the rebuttal, the authors address my concerns by adding results on two larger datasets. Moreover, they discuss the difference between Prop.3 and Theorem 2 in [5]. Finally, they add a discussion of the convergence rate analysis of the proposed method.
> >
> > Thank you for providing your response and additional clarifications.

---

### Official Review · Reviewer_MUdC · 2024-07-12

**Soundness:** 3
**Presentation:** 3
**Contribution:** 2
**Rating:** 6
**Confidence:** 3

**Summary:**

This manuscript deals with the **Normlaized cut (NCut)** problem by proposing a new algorithm called **fractional programming-based clustering** (FPC). The main idea of FPC is rewriting the original NCut problem to an equivalent one by using a so-called **Multidimensional quadratic transform**. Then, the clustering result is obtained by an iterative step that is guaranteed to monotonically increase the objective of NCut problem until it converges. Experiments demonstrate that the proposed algorithm outperforms the baselines with a higher objetive value on multiple datasets.

**Strengths:**

I think the strengh of this paper mainly falls onto the following aspects:

**Novelty**:
- Rewriting the NCut problem by the Multidimensioanl quadratic transform
- A critical step (Eq. (23) in the manuscript) is proposed to make the clustering result $\boldsymbol{X}$ can be efficiently solved
- The designed FPC finds the clustering iteratively that monotonically increases the objective of NCut

**Experimental result**
- The proposed FPC outperforms other baselines with a higher objective value of NCut obtained on multiple real datasets
- It also gives a better clustering result in terms of metrics such as accuracy and normalized mutual information
- The runtime of FPC seems to be similar to the baselines

**Clarity**
- The writing of this manuscript is clear and easy to follow
- The proofs are complete that necessarily supports the proposed arguments

Overall, I like the idea of this manuscript and I think the novelty is sufficient to NeurIPS. The proposed algorithm should be a good supply to the field of graph cut that may inspires some further research under the same direction, as long as several concerns (see the weaknesses below) can be adequately addressed.

**Weaknesses:**

**(1) Lack of analysis and experiments on the convergence of FPC**

Since it is guaranteed that the objective value of NCut is monotonically increasing during the iterations, it is very important to understand how does the convergence of FPC look like. For example, I would like to know **if the objective value could get stuck on some sub-optimal point?** And if yes, **when does this happen in practice?** Without any study (either theoretical or emprical) on the convergece, it is hard to tell if the algorithm has reached its limit or not. So I would suggest the authors to at least show some curves of the objective values during the iterations.

**(2) How far to the global optimum?**

Even though the objective values obtained from FPC are higher than other baselines' in the experiments, it is still unknown how important are such improvements compared to the global optimum. So I would be curious on some cases (could be small and artificial) when global optimum are known and see how far is the result of FPC to the global optimum. So far, the objective value by itself is not that meaningful to me and may not be able to fully showcase the strength of the algorithm

**Questions:**

My questions are mainly described in the Weaknesses section.

Besides, I wonder if the author could conduct some experiments that compares the performance of the algorithm (especially on the aspect of convergence) on clustering scenarios with **difficulties from easy to hard**, i.e. simply construct a Gaussian Mixtures with two clusters and gradually adjust their center distance from distant to close, and see during the transition, if the convergence becomes more challanging. This could be helpful to understand the limits of the algorithm.

**Limitations:**

Yes

---

> ### Author Rebuttal · Authors · 2024-08-06
>
> Thank you so much for the very positive comments on this work. Also thank you so much for providing many constructive suggestions. We have added the convergence rate analysis and many new experiments according to your suggestions, as specified in what follows.
>
> 1. **Weakness One:** We would edit both the theoretic and the experimental aspects of this paper significantly to relieve your concern. On the theory side, we accomplish the convergence rate analysis of the proposed FPC algorithm. It can be shown that
> \begin{align}
> |f(x^*)-f(x^{(1)})|  \le \frac{LR^3}{6},
> \end{align}
> \begin{align}
> |f(x^*)-f(x^{(k)})|  \le \frac{2\Lambda R^2+2LR^3/3}{k+3},\quad\text{for}\quad k\ge 2
> \end{align}
> where $f(\cdot)$ is the optimization objective value, $x^*$ is the converged solution,
>     $k$ is the iterate index, $R$ is the Euclidean distance from the starting point to $x^*$, $L$ is the Lipschitz constant of $\nabla^2 f(x)$, $\Lambda$ is the maximum eigenvalue of $\nabla^2 f(x)$, and $x^{(k)}$ is the solution after $k$ iterates. We remark that the convergence rate analysis is highly nontrivial here because the NCut problem is nonconvex and incurs discrete constraints. On the experiment side, we add new numerical results on the convergence behavior of the proposed algorithm. In the one-page PDF, Figure 1 shows how fast the FPC algorithm converges; observe that FPC attains convergence after merely 3 iterates. Moreover, Figure 2 shows the local optimum issue of FPC. When two cluster centers are far apart, FPC can always achieve the global optimum; but when the cluster centers get closer so that the NCut problem becomes more difficult (as advised by the reviewer), then FPC may get stuck at a local optimum. In practice, we may reduce the risk of local optimum trapping by trying out various starting points.
> 2. **Weakness Two:** We have incorporated this excellent advice into our work. Specifically, we find the global optimum for those small-size dataset via exhaustive search, and use it as the benchmark to compare with the proposed FPC algorithm. The new results are shown in Figure 1 in the attached one-page PDF. The figure shows that the FPC attains the global optimum.
> 3. **Questions:** Thank you for the nice advice, which has been fully implemented. As the reviewer suggests, we try out the FPC algorithm on the different clustering scenarios with the distance between two cluster centers being gradually reduced, and consequently the clustering problem becomes increasingly difficult. This new experiment is shown in Figure 2 in the attached one-page PDF, which will be incorporated into the paper after the rebuttal session finishes.

---

> > ### Comment · Reviewer_MUdC · 2024-08-13
> > **Response to rebuttal**
> >
> > Thank the authors for the detailed explanations, and most of my concerns and questions have been addressed in the reponse. Specifically, the toy example on a two-Gaussian Mixture clearly demonstrates the performance of the proposed algorithm, and also reveals the potential issue of being trapped at local optimums. It would be a good complement to the current analysis of this manuscript.
> >
> > Overall, I would keep my rating and thanks again for the response.

---

### Official Review · Reviewer_mC6e · 2024-07-13

**Soundness:** 2
**Presentation:** 2
**Contribution:** 2
**Rating:** 3
**Confidence:** 4

**Summary:**

The paper addresses the challenge of the Normalized Cut (NCut) problem in unsupervised clustering.

Conventional fractional programming (FP) techniques, especially Dinkelbach’s transform, are inadequate as they only handle single ratios and are limited to two-class clustering.

This paper extends the quadratic transform to multidimensional ratios, converting the fractional 0-1 NCut problem into a solvable problem.

The authors also show the convergence of their proposed multidimensional FP method using minorization-maximization theory.

**Strengths:**

1. The proposed method extends the quadratic transform to handle multiple ratios, enabling it to address multi-class classification problems.

2. The proposed method converts the complex NCut problem into a more tractable form, solving it iteratively as a manageable subproblem.

3. The algorithm's performance is validated on multiple datasets, demonstrating superior results compared to existing methods.

**Weaknesses:**

1. **Application Reformulation**: Reformulating the NCut problem into a multiple ratio problem is not new and has been previously demonstrated in the literature.

2. **Multiple Ratio Fractional Optimization**: Solving multiple ratio fractional optimization using the Quadratic Transform is a standard practice. The convergence of the proposed method follows directly from existing minorization-maximization techniques. The proposed algorithm and its associated theoretical analysis are relatively incremental.

Overall, the problem formulation and optimization algorithm in this paper have limited novelties.

**Questions:**

What are the main novelties of this paper?

**Limitations:**

Yes

---

> ### Author Rebuttal · Authors · 2024-08-06
>
> Thank the reviewer for acknowledging the strengths of this paper. We would like to focus on the "Weakness'' and "Questions'' parts in the following.
>
> 1. **Weakness One:** Actually, the paper never claims that the reformulation  of the NCut problem as a multiple-ratio problem is a contribution or novelty. The real contribution lies in how to address the 0-1 NCut problem and the corresponding performance analysis, as opposed to the previous works that simply drop the discrete constraint in a heuristic fashion.
> 2. **Weakness Two:** It is true that our method is more or less connected to the existing fractional programming theory. But we wish to clarify that (almost) all the optimization algorithms in the AI field are based on the existing optimization methods/theories, e.g., the well-known Adam optimizer is in essence a momentum-aided gradient method---which has been extensively studied in the optimization community before Adam was proposed. Moreover, we wish to highlight the fundamental difference between our work and the existing literature. There are several highly nontrivial new results/insights that are by no means incremental improvements, e.g., the comparison between the scalar-ratio method and the matrix-ratio method, the new quadratic transform method tailored to the NCut problem, and the convergence rate analysis.
> 3. **Conclusion of Weakness:** Again, first of all, we never ever claim the problem formulation as a novelty. This work is aimed at solving a notoriously difficult long-standing problem from a novel perspective. Secondly, this work is much more than just applying the standard quadratic transform to the NCut problem; we have provided many highly nontrivial new results/insights.
> 4. **Questions:** The main novelties of this paper can be recognized in the following three respects:
>     -  *New Method:* Our main theoretic contribution is stated in Proposition 3. We clarify that it is fundamentally different from the existing quadratic transform in Theorem 2 in [5]. Actually, the conventional quadratic transform in the literature does not even work for the NCut problem. The new fractional programming method considerably generalizes the existing one to account for a wider range of multiple-ratio problems.
>     -  *New Insight:* There are two types of quadratic transform: the scalar case and the matrix case. In the literature, the choice of quadratic transform simply depends on the original form of the ratios contained in the problem, i.e., the scalar (resp. matrix) quadratic transform is employed if the ratios are scalars (resp. matrices). However, this paper shows that, even though the ratios in the NCut problem are scalar, it is better to rewrite the scalar ratios in the matrix form and thereby apply the matrix quadratic transform, otherwise the discrete constraint is difficult to tackle. This is an interesting nontrivial insight.
>     -  *New Analysis:* Although the convergence behavior of the quadratic transform has been studied extensively in the literature, most of the prior works only discuss the conditions under which the quadratic transform based iterative algorithm can guarantee convergence. To the best of our knowledge, it remains a mystery as to how fast the quadratic transform method converges. As a newly added result, we show that
>    \begin{align}
>         |f(x^*)-f(x^{(1)})| \le \frac{LR^3}{6},
>     \end{align}
> \begin{align}
> |f(x^*)-f(x^{(k)})| \le \frac{2\Lambda R^2+2LR^3/3}{k+3},\quad\text{for}\quad k\ge 2,
> \end{align}
>    where $f(\cdot)$ is the optimization objective value, $x^*$ is the converged solution,
>     $k$ is the iterate index, $R$ is the Euclidean distance from the starting point to $x^*$, $L$ is the Lipschitz constant of $\nabla^2 f(x)$, $\Lambda$ is the maximum eigenvalue of $\nabla^2 f(x)$,
>     and $x^{(k)}$ is the solution after $k$ iterates. We emphasize that the convergence rate analysis is highly nontrivial due to the nonconvexity and the discrete constraint of the NCut problem.

---

> > ### Comment · Reviewer_mC6e · 2024-08-10
> >
> > 1. First, as the authors acknowledge, this paper does not claim the problem formulation as a novel contribution. Now, let's discuss the multiple ratio fractional problem, specifically Problem (7) in the paper.
> >
> > 2. Using the quadratic transform $\frac{g(X)}{f(X)} = \max_{\beta} \left(2 \beta \sqrt{g(X)} - \beta^2 f(X)\right)$, the authors transform the original multiple-ratio problem $\max_{X} \sum_{i} \frac{g_i(X)}{f_i(X)}$ into an optimization problem involving two blocks:
> >
> > $~~~~~~\max_{X} \max_{\beta} \sum_{i} \left(2 \beta_i \sqrt{g_i(X)} - \beta_i^2 f_i(X)\right), X \in \Omega.$
> >
> > $\quad$ Here, $ \Omega $ is the constraint set in (7b) and (7c) in the perper.
> >
> > $\quad$ This is a very standard practice in solving multiple ratio fractional optimization.
> >
> > 3. For the variable $\beta$, the authors use the standard quadratic transform update rule. For the variable $X$, since the constraints are non-convex and have a discrete structure, an additional term $-||X||_F^2$ can be added to the objective function to make it strongly concave with respect to $X$ (which is equivalent to adding a constant).  Consequently, methods like the conditional gradient method or power method can be employed to maximize over $X$. By alternating between maximization over X and $\beta$, the algorithm can be shown to be monotonically increasing. The alternating maximization strategy is also a very standard approach [R2,R3].
> >
> > 4. Although the authors extend this to matrix methods in "Section 3.2 Multidimensional FP method," they do not discuss the specific motivation, practical applications, or experimental results. Moreover, matrix methods have already been explored in the literature [R3].
> >
> > 5. The authors reply that: "The new fractional programming method considerably generalizes the existing one to account for a wider range of multiple-ratio problems." This is incorrect. For problem (7), where both the numerator and denominator are quadratic and the constraints are straightforward, there are already established algorithms [R1,R2,R3]. The quadratic transform was originally proposed to handle the optimization of **multiple-ratio problems**, and naturally, problem (7) falls within its scope.
> >
> > 6. The theoretical contributions of this paper are quite limited. The authors only establish a sufficient descent property for the algorithm, which can be directly derived using the classical Majorization Minimization technique. Although the authors claim to have added new analysis to the paper, I do not find such results.
> >
> > 7. The proposed algorithm is merely another heuristic approach. While it converges to a fixed point, it lacks any theoretical guarantee of optimality. Furthermore, there is no intuitive or theoretical justification for why it leads to improved experimental results over existing methods.
> >
> > [R1] XueGang Zhou and JiHui Yang. Global optimization for the sum of concave-convex ratios problem. Journal of Applied Mathematics, 2014.
> >
> > [R2] Radu Ioan Bot, Minh N. Dao, Guoyin Li. Inertial Proximal Block Coordinate Method for a Class of Nonsmooth Sum-of-Ratios Optimization Problems. SIOPT 2023.
> >
> > [R3] K. Shen and W. Yu, Fractional programming for communication systems–Part I: Power control and beamforming, IEEE Trans. Signal Process 2018.

---

> > > ### Author Response · Authors · 2024-08-11
> > > **Our response to new comments of Reviewer mC6e**
> > >
> > > Thanks so much for the further comments and for giving us this opportunity to clarify. To avoid any possible confusion, we now refer to our proposed method in Prop. 3 as MQT (matrix quadratic transform), the existing method in Theorem 1 in [R3] as SQT (scalar quadratic transform), and the existing method in Theorem 2 in [R3] as VQT (vector quadratic transform). We would like to answer your detailed comments in the following.
> > > 1. Thanks for accepting our previous argument.
> > > 2. Sorry but the method you describe here is NOT our method. The method you refer to is SQT, while our method is MQT. We will show that SQT does NOT work for the NCut problem later.
> > > 3. Again, we are not applying SQT to the NCut problem as the reviewer thinks; we are using MQT. The gradient method suggested by the reviewer is typically limited to the continuous optimization, which does not account for the discrete constraint directly, so it has to relax the discrete variables to be continuous and consequently its performance becomes unpredictable.
> > > 4. This is a big misunderstanding! We beg to differ. The new matrix method in Prop. 3, namely MQT, is the building block of this work. Now let us show why the existing standard methods, SQT and VQT, do not work for the NCut problem. Recall that the NCut problem is
> > > \begin{equation}
> > >         \begin{aligned}
> > >             \underset{X}{\text{maximize}}&\quad \sum_{k=1}^K\frac{x_k^\top W x_k}{x_k^\top Dx_k}\\\\
> > >      \text{subject to}&\quad \sum_{k=1}^K X_{ik} = 1,\quad\forall i\\\\
> > >             &\quad X_{ik}\in\\{0,1\\}, \quad \forall i,\forall k,
> > >         \end{aligned}
> > >     \end{equation}
> > > where $x_k=[X_{1k},X_{2k},\ldots,X_{Nk}]^\top$. If we apply SQT then the new problem is
> > > \begin{equation}
> > >     \begin{aligned}
> > >         \underset{X,y_k\in\mathbb R}{\text{maximize}}&\quad \sum_{k=1}^K\left(2y_k\sqrt{x_k^\top Wx_k}-y^2_k{x_k^\top Dx_k}\right)\\\\
> > >      \text{subject to}&\quad \sum_{k=1}^K X_{ik} = 1,\quad\forall i\\\\
> > >             &\quad X_{ik}\in\\{0,1\\}, \quad \forall i,\forall k,
> > >     \end{aligned}
> > > \end{equation}
> > >  However, the new problem is still a nonlinear integer program, so the optimization for $X_{ik}\in\\{0,1\\}$ remains quite difficult.
> > >  We further show that VQT in Theorem 2 in [R3] does not work for the NCut problem either. Recall that by the VQT, the vector ratio problem
> > >
> > >     $$\underset{X\in\mathcal X}{\text{maximize}}\quad\sum_{k=1}^K a_k^\top(X) B^{-1}_k(X) a_k(X)$$
> > >
> > >     with $a_k(X)\in\mathbb R^d$ and $B_k(X)\in\mathbb S^{d\times d}$
> > >     is recast to
> > >
> > >     $$\underset{X\in\mathcal X,y_k\in\mathbb R^d}{\text{maximize}}\quad\sum_{k=1}^K y_k^\top a_k(X)-y_k^\top B^{-1}_k(X)y_k.$$
> > >
> > >       For the NCut problem, each $x_k^\top Dx_k \in \mathbb{R}$ is treated as $B_k(X)$, so we end up with $d=1$ and $a_k(X) = \sqrt{X_k^\top W X_k}$. As a result, VQT leads to the same reformulation as SQT in the NCut problem case, so it also cannot render the integer variable $X_{ik}\in\\{0,1\\}$ easier to tackle.
> > >
> > >      In contrast, MQT aims at a more general matrix ratio problem
> > >         $$\underset{X\in\mathcal X}{\text{maximzie}}\quad \sum_{k=1}^n\mathrm{tr}\left(B_k^{-1}(X)A_k(X)\right),$$
> > >     where $B_k(X)\in\mathbb S_{++}^{d\times d}$ and $A_k(X)\in\mathbb S_{+}^{d\times d}$, and recasts it to
> > >    $$\underset{X\in\mathcal X,Y_k\in\mathbb{R}^{\ell\times d}}{\text{maximize}}\quad\sum_{k=1}^n \mathrm{tr}\left(2Y_k[Z_k(X)]^\top-Y_k B_k(X)Y_k^\top\right),$$
> > >     where
> > >       $$A_k(X) = [Z_k(X)]^\top[Z_k(X)]\quad\text{for some}\quad Z_k(X)\in\mathbb R^{\ell\times d}$$
> > >     for the given positive integer $\ell\ge1$. Observe that MQT reduces to VQT when $\ell=1$.
> > >
> > >     In light of MQT, the NCut problem (1) is converted to
> > >     \begin{equation}
> > >         \begin{aligned}
> > >             \underset{X,y_k\in\mathbb{R}^{N}}{\text{maximize}}&\quad \sum_{k=1}^K \left(2y_k^\top{W}^{\frac12}x_k-y_k^\top y_k\delta^\top x_k\right)\\\\
> > >      \text{subject to}&\quad \sum_{k=1}^K X_{ik} = 1,\quad\forall i\\\\
> > >             &\quad X_{ik}\in\\{0,1\\}, \quad \forall i,\forall k,
> > >         \end{aligned}
> > >     \end{equation}
> > >     where $\delta=1^\top D$.
> > >     Now we arrive at a linear integer problem which can be immediately solved by the standard matching method.
> > >
> > >     Due to the length limit, we would like to answer your questions 5 to 7 in a separate response that follows the current one.

---

> ### Author Response · Authors · 2024-08-11
> **Our response to new comments of Reviewer mC6e (Cont.)**
>
> Sorry about the break. Now please allow us to continue to answer your questions 5 to 7.
>
> 5. Let us reiterate the distinctions between the various methods. SQT and VQT can be found in the existing literature [R1, R2, R3], while MQT is what we newly propose. We remark that:
>    - SQT considers the sum-of-scalar-ratios problem:
>         $$\text{maximize}\quad\sum_{i=1}^K \frac{A_i}{B_i},$$
>
>
>         where each $A_i\in \mathbb R$, each $B_i\in\mathbb R$, and each ratio
>         $$
>         \frac{A_i}{B_i}\in\mathbb R.
>         $$
>    - VQT considers a generalized problem:
>         $$\text{maximize}\quad\sum_{i=1}^K a^\top_i B^{-1}_i  a_i$$
>         where each $a_i\in\mathbb R^{d}$, each $B_i\in\mathbb S^{d\times d}$, and each ratio
>         $$
>          a_i^\top B_i^{-1}  a_i \in\mathbb R.
>         $$
>    - MQT considers a further generalized problem:
>         $$\text{maximize}\quad\sum_{i=1}^K \text{tr}(B^{-1}_i A_i),$$
>
>         where each $A_i\in\mathbb S^{d\times d}$, each $B_i\in\mathbb S^{d\times d}$, and each ratio
>      $$B_i^{-1}A_i \in\mathbb R^{d\times d}.$$
>      In particular, we assume that each $A_i$ can be factorized as
>          $$A_{i} = Z_i^\top Z_i \text{   for some  } Z_i\in\mathbb R^{\ell \times d}.$$
>         Note that when the ratio is nested in $\mathrm{Tr}(\cdot)$, we can rewrite the VQT problem  as the MQT problem:
>         $$\mathrm{Tr}( a_i^\top B_i^{-1}  a_i)=\mathrm{Tr}( B_i^{-1}  a_i a_i^\top)=\mathrm{Tr}( B_i^{-1}  A_i), \text{where}   A_i =  a_i a_i^\top.$$
>      However, the reverse is false since not every matrix $A_i$ can be factored as an outer product $a_i a^\top_i$. Thus, MQT is strictly more general than VQT, while VQT is strictly more general than SQT.
> 6. The theoretical contributions of this paper
>     are two-fold aside from the MM interpretation. First, as we repeatedly emphasize, the proposed MQT in Prop. 3 is a brand-new FP method, which strictly generalizes the existing SQT and VQT. Second, we analyze the convergence rate of MQT. Due to the length limit, we only provide a sketched proof of this new result in the following.
> Denote by $x$ the vectorization of $X$. To ease analysis, we now write the NCut objective function as a function of $ x$.Conditioned on $ x'\in\mathcal X$, write the difference between the original NCut objective function $f(x)$ and the new objective function $h(x,y)$ by MQT as a function of $ x\in\mathcal X$:
> $$\delta(x|x') = f(x) - h(x,\mathcal Y(x')),$$
> where $\mathcal Y(x')$ refers the optimal update of each $Y_k$ based on current $x'$. Moreover, define the following quantity
> $$\Lambda = \sup_{x\in\mathcal X} \lambda_{\max}\big(\nabla^2\delta(x|x')\big)=\sup_{x\in\mathcal X} \lambda_{\max}\big(\nabla^2f(x)\big),$$
> where $\lambda_{\max}(\cdot)$ is the largest eigenvalue of the given matrix. With the iteration index denoted by $t$, we can bound the cubic Euclidean norm as
> \begin{align*}
> &\frac{L}{6}\|x- x^{t-1}\|^3_2\notag\\\\
> &\ge \delta(x|x^{t-1})-\frac{\Lambda}{2}\|x-x^{t-1}\|^2_2\notag\\\\
> &= f(x)-h(x,y^t)-\frac{\Lambda}{2}\|x-x^{t-1}\|^2_2\notag\\\\
> &\overset{(a)}{\ge} f(x)-h(x^t,y^t)-\frac{\Lambda}{2}\| x- x^{t-1}\|^2_2\notag\\\\
> &\overset{(b)}{\ge} f(x)-h( x^t, y^{t+1})-\frac{\Lambda}{2}\|x-x^{t-1}\|^2_2\notag\\\\
> &= f(x)-f(x^t)-\frac{\Lambda}{2}\|x-x^{t-1}\|^2_2,
> \end{align*}
> where step $(a)$ follows since $x^t$ maximizes $h(x, y)$ for the current $y=y^t$, and step $(b)$ follows since $ y^{t+1}$ maximizes $h(x,y)$ for the current $x=x^t$. Furthermore, denote the gap in the objective value as
>    $$ v_t = f(x^*)-f(x^t),$$
> which can be bounded from above as
> $$v_t
> \le (1-\pi) v_{t-1}+\frac{\pi^2\Lambda }{2}\| x^*- x^{t-1}\|^2_2+\frac{\pi^3L}{6}\| x^*- x^{t-1}\|^3_2$$
>
>      $$\quad\le (1-\pi) v_{t-1}+\pi^2\bigg(\frac{\Lambda R^2}{2}+\frac{LR^3}{6}\bigg),\tag{1}$$
>
>     where $R$ is the Euclidean distance from the starting point to $x^*$.  When $t=1$, we let $\pi=1$ in (1) and obtain
>      $$v_1\le \frac{\Lambda R^2}{2}+\frac{LR^3}{6}.\tag{2}$$
>
>     When $t\ge2$, we let
>     $$\pi = \frac{v_{t-1}}{\Lambda R^2+LR^3/3}.$$
>
>     Then after some algebra, we ultimately obtain
>     \begin{align*}
>     \frac{1}{v_t}
>     &\ge \frac{1}{v_1} + \frac{t-1}{2\Lambda R^2+2LR^3/3}\notag\\\\
>     &\ge \frac{t+3}{2\Lambda R^2+2LR^3/3},
>     \end{align*}
>     where the second inequality follows by (2). The proof is then completed.
> 7. Since the NCut problem is NP-complete as shown in Ref. [4], one has to resort to the branch-and-bound algorithm as in [R1] to guarantee optimality, but it has exponential complexity and thus is not suited for large dataset. The other reference [R2] recommended by the reviewer does not provide any optimality guarantee. Our method is much more than a heuristic for three reasons: (i) we connect it to the MM theory and thus all the desirable properties of MM carry over to it; (ii) we generalize the existing SQT and VQT to MQT, which plays a crucial role in solving the nonlinear integer program of NCut; (iii) we provide convergence rate analysis.

---

### Author Rebuttal · Authors · 2024-08-06

First of all, we wish to thank the TPC members for organizing reviews for our paper. The comments from Reviewer MUdC and Reviewer fiXH are quite positive; they both think that the paper is well written and contains sufficient novelty and technical contributions in terms of the NeurIPS criterion. The two reviewers provide some constructive suggestions (e.g., add convergence analysis and new experiments, and use larger datasets), all of which have been accomplished and can be readily incorporated into the paper.


In contrast, Reviewer mC6e has expressed deep concerns about the novelty of this work. But we believe that this is due to misunderstanding. Reviewer mC6e criticizes that formulating the NCut problem as a multiple-ratio problem should not count a novelty. However, this paper never claims this problem formulation as any sort of novelty/contribution. Our real contribution lies in solving this notoriously difficult long-standing problem from a novel fractional programming perspective, and also in highly nontrivial performance analysis.

The other criticism from Reviewer mC6e concerns the novelty of the proposed method. He or she thinks that the proposed FPC algorithm just follows the existing quadratic transform method in [5]. But the technical contributions of this paper are much more than that:
- As we have confirmed with Reviewer mC6e and Reviewer fiXH, our main theoretic contribution stated in Proposition 3 is fundamentally different from the existing quadratic transform in Theorem 2 in [5]. Actually, the conventional quadratic transform in the literature does not even work for the NCut problem. The new fractional programming method considerably generalizes the existing one to account for a wider range of multiple-ratio problems.
- We tailor quadratic transform to the NCut problem and bring new insight. There are two types of quadratic transform: the scalar case and the matrix case. In the literature, the choice of quadratic transform simply depends on the original form of the ratios contained in the problem, i.e., the scalar (resp. matrix) quadratic transform is employed if the ratios are scalars (resp. matrices). However, this paper shows that, even though the ratios in the NCut problem are scalar, it is better to rewrite the scalar ratios in the matrix form and thereby apply the matrix quadratic transform, otherwise the discrete constraint is difficult to tackle.
- The convergence rate analysis is by no means an incremental improvement upon the previous works. Although the convergence behavior of the quadratic transform has been studied extensively in the literature, most of the prior works only discuss the conditions under which the quadratic transform based iterative algorithm can guarantee convergence. To the best of our knowledge, it remains a mystery as to how fast the quadratic transform method converges. As a newly added result, we show that

\begin{align}
        |f(x^*)-f(x^{(1)})| \le \frac{LR^3}{6},
\end{align}

\begin{align}
        |f(x^*)-f(x^{(k)})| \le \frac{2\Lambda R^2+2LR^3/3}{k+3},\quad\text{for}\quad k\ge 2,
    \end{align}
where $f(\cdot)$ is the optimization objective value, $x^*$ is the converged solution, $k$ is the iterate index, $R$ is the Euclidean distance from the starting point to $x^*$, $L$ is the Lipschitz constant of $\nabla^2 f(x)$, $\Lambda$ is the maximum eigenvalue of $\nabla^2 f(x)$,
and $x^{(k)}$ is the solution after $k$ iterates. We emphasize that the convergence rate analysis is highly nontrivial due to the nonconvexity and the discrete constraint of the NCut problem.

The above key contributions and novelties may have been overlooked in the last round of review. We sincerely hope that our responses can help highlight them.

---

### Decision · Program_Chairs · 2024-09-25

**Decision:**

Accept (poster)

**Comment:**

The authors proposed to solve Normalized cut via the fractional programming technic. Both Normalized cut and fractional programming are not new, but the usage of fractional programming is new. The experiments need to be improved